# Expanding the fluorescent toolkit: Blue fluorescent protein-expressing *Plasmodium berghei* for enhanced multiplex microscopy

Kodzo Atchou[1,2], Reto Caldelari[1], Magali Roques[1], Jacqueline Schmuckli-Maurer[1], Raphael Beyeler[1,2], Volker Heussler[1] *

1 Institute of Cell Biology, University of Bern, Bern, Switzerland, 2 Graduate School for Cellular and Biomedical Sciences, University of Bern, Bern, Switzerland

* volker.heussler@unibe.ch

**Data Availability Statement:** All data generated or analyzed during this study are included in this

## Abstract

Fluorescent proteins are widely used as markers to differentiate genetically modified cells from their wild-type counterparts. In malaria research, the prevalent fluorescent markers include red fluorescent proteins (RFPs) and their derivatives, such as mCherry, along with green fluorescent proteins (GFPs) and their derivatives. Recognizing the need for additional fluorescent markers to facilitate multiplexed imaging, this study introduced parasite lines expressing blue fluorescent protein (BFP). These lines enable simultaneous microscopy studies of proteins tagged with GFP, RFP, or detected by fluorophore-labeled antibodies, enhancing the analysis of complex biological interactions. Expression of BFP throughout the parasite's life cycle was driven by the robust Hsp70 promoter, ensuring stable, detectable protein levels suitable for fluorescent light analysis methods, including flow cytometry and fluorescent microscopy. We generated two *Plasmodium berghei* (*P. berghei*) lines expressing cytosolic BFP through double crossover homologous recombination targeting the silent 230p locus: eBFP2 (PbeBFP2) and mTagBFP2 (PbmTagBFP2). We compared these transgenic lines to established mCherry-expressing parasites PbmCherry$_{Hsp70}$ (PbmCherry) across their life cycles. The PbmTagBFP2 parasites exhibited fluorescence approximately 4.5 times brighter than the PbeBFP2 parasites in most life cycle stages. Both BFP-expressing lines developed normally through the entire parasite life cycle, offering a valuable expansion to the toolkit for studying *Plasmodium* biology at the host-pathogen interface.

## Introduction

Rodent malaria models are of eminent importance for malaria research as they share similar biology with the human *Plasmodium* species. *Plasmodium* species infecting rodents include *P. berghei*, *P. chabaudi*, *P. vinckei, and P. yoelii* [1]. Genetic manipulations, such as the generation of gene-knockout and protein-tagged parasite lines, are possible for both rodent and human *Plasmodium* species [2–6]. The *Plasmodium* life cycle is complex, alternating between a

published article and its Supplementary Information files.

**Funding:** The authors declare that this research was conducted without any commercial or financial relationships that could be construed as a potential conflict of interest. This study was funded by the Swiss National Science Foundation (SNSF) (grant number 310030_212795) and the Multidisciplinary Center for Infectious Diseases (MCID) grant MA-09 to Volker Heussler and by the Swiss Confederation Government Excellence Scholarship to Kodzo Atchou. The funders had no role in study design, data collection and analysis, decision to publish, or preparation of the manuscript. All authors confirm that there are no financial conflicts of interest to disclose.

**Competing interests:** The authors declare no competing interests.

mosquito and a vertebrate host. The parasite is transmitted to the mammalian host by an infected female *Anopheles* mosquito. During the mosquito's blood meal, up to a few hundred *Plasmodium* sporozoites are inoculated into the host skin. About a third of these sporozoites invade blood vessels, where they are passively carried to the liver. The remaining sporozoites are either taken up and eliminated by dendritic cells in the skin or invade lymph vessels and are eliminated in the draining lymph nodes [7]. Parasites that reach the liver transmigrate through the blood vessel epithelium and productively invade hepatocytes, where they reside in a parasitophorous vacuole formed by the invagination of the host cell plasma membrane. After some initial transformations, the sporozoites develop into trophozoites and then multiply their nuclei in a process called schizogony, resulting in the formation of tens of thousands of merozoites, depending on the *Plasmodium* species. Merozoites leave the liver parenchyma and enter the blood vessels inside merosomes, membranous structures of host cell origin that protect them until they are released into the bloodstream. Free merozoites invade erythrocytes, where they develop and multiply to form new merozoites. Some of the merozoites commit to becoming sexual forms, gametocytes, which are taken up by a female mosquito during a blood meal and activated to form gametes in the mosquito midgut. Male and female gametes then fuse and form a zygote, from which the motile ookinete develops. Upon traversing the midgut, the ookinetes develop into oocysts between the basal membrane of the midgut and the basal lamina. Thousands of sporozoites are formed within the oocyst and are released into the mosquito hemocoel to eventually reach and penetrate the salivary gland [8]. Despite a relatively good understanding of the overall parasite's life cycle, many molecular and cellular processes and parasite-host interactions remain to be discovered to find new antimalarial agents [9–11]. One challenging key in malaria research is to dissect the molecular details of the different life cycle stages of the parasite, which is limited by the availability of reporter parasite lines. In other cell systems and model organisms, a diverse spectrum of fluorescent proteins has been established for multiplexed imaging of distinct cellular components [12–15]. However, besides the excitation and emission wavelengths, other characteristics, such as pH and photostability, of the fluorescent proteins, are important. Furthermore, phototoxicity can pose an additional challenge that must be considered for live cell imaging [16, 17]. Technological developments in the field of microscopy, combined with the generation of both fluorescent parasites and mouse models with specific cell types expressing fluorescent proteins, have allowed for cutting-edge live cell and intravital imaging of the entire *Plasmodium* life cycle [18, 19]. However, the repertoire of available parasites expressing different fluorescent proteins remains limited in human malaria parasites like *P. falciparum* and the rodent model parasite *P. berghei*. Although some transgenic *P. falciparum* and *P. berghei* lines expressing GFP and RFP have been successfully developed and utilized in various studies [20–25] there is still a gap in the availability of lines expressing alternative fluorescent proteins, such as blue fluorescent proteins (BFPs). The development of these tools for *P. falciparum* and *P. berghei* could offer significant benefits for studying complex interactions at the host-parasite interface, especially in detailed imaging applications where spectral separation is critical. Transgenic *P. falciparum* and *P. berghei* lines expressing BFP could enable simultaneous imaging alongside existing red and green fluorescent lines, facilitating the study of multi-protein complexes, parasite organelle dynamics, and host cell responses during different stages of the parasite's life cycle.

The aim of this study was to develop transgenic *P. berghei* parasites expressing BFP for multiplexed imaging throughout the parasite life cycle as well as to study host-pathogen interactions. Two BFPs with different features were chosen for this study: mTagBFP2, which is fluorescent as a monomeric molecule, and eBFP2, which requires dimerization to fluoresce. The use of mCherry-transgenic parasites is limited for multiplexing as the combination with far-red and green fluorophores often results in bleaching of the fluorescent signals during live

**Table 1. Overview of mCherry, eBFP2 and mTagBFP2 characteristics [16, 26].**

| Fluorescent proteins | Molecular weight | Excitation maximum, nm | Emission maximum, nm | Extinction coefficient, $M^{-1}$ $cm^{-1}$ | Brightness | Fluorescence lifetime, ns | Effective p$Ka$ | Photobleaching $t_{1/2}$, sec |
|---|---|---|---|---|---|---|---|---|
| mCherry | 26.7 kDa | 587 | 610 | 72000 | 15840 | 1,4 | 4,5 | 68 |
| eBFP2 | 26.9 kDa | 383 | 448 | 32000 | 17920 | 3,0 | 5.3 | 55 |
| mTagBFP2 | 26.7 kDa | 399 | 454 | 50600 | 32384 | 2,6 | 2,7 | 53 |

imaging. The narrow emission peak of BFP allows for better spectral separation. Additional advantages of BFPs include improved photostability and the low pKa, which increases stability under low pH and enables imaging of the parasites even in acidic cellular compartments, for example when they are eliminated in host cell lysosomes or autophagosomes (Table 1) [16, 26–28].

## Materials and methods

### Animal work statement

Experiments were performed strictly according to the Swiss Act on animal protection (TSchG; Animal Rights Laws). They were approved by the animal experimentation commission of the canton Bern (Permit Numbers BE86/19 and BE118/22). The research staff received their training at Certified education and training programs from the Federal European Laboratory Animal Science Associations (FELASA). The mice (C57BL/6 and BALB/c) were between six and ten weeks old and were either bred in-house or supplied from Janvier Labs (France). The mice were monitored daily in the mice facility of the Institute of Cell Biology at the University of Bern.

### Transgenic eBFP2 and TagBFP2 parasite lines

To generate the BFP-reporter cassettes, the previously generated mCherry-reporter cassette pL1694 was used [29, 30]. The pL1694 construct contains the *5'hsp70-mCherry-3'hsp70* cassette, which allows mCherry expression under the heat shock protein 70 (Hsp70) promoter. The mCherry was removed using BamH1/Not1 and replaced by polymerase chain reaction (PCR)-amplified eBFP2 to generate the *5'hsp70-eBFP2-3'hsp70* cassette. The same approach was used to generate the *5'hsp70-mTagBFP2-3'hsp70* cassette. The eBFP2 was amplified from the pEBFP2-Nuc (Addgene #14893), and mTagBFP2 from pHAGE-BL-mTagBFP2-SopF from Richard Youle Laboratory [31]. The eBFP2 and mTagBFP2 were amplified using primers as indicated in Table 2 and subcloned into pJet1.2/blunt. The vectors were transfected into the GIMO (Gene Insertion/Marker Out) ANKA parasite mother line (GIMO$_{PbANKA}$,1596cl1, RMgm-687) as described [29].

### Transgenic parasites transfection and selection

To transfect the eBFP2 and mTagBFP2 vectors into the GIMO ANKA parasite line, a schizont culture approach was used as previously described [29, 32–34]. A mouse was pre-infected intraperitoneally (IP) with the *P. berghei* GIMO mother line. The blood of the infected mouse was intravenously (IV) passaged three days later into a new naive mouse. When the passaged mouse's parasitemia reached 1–3%, the schizont culture was prepared. For every transfection, a schizont culture medium was produced using 50 mL of Roswell Park Memorial Institute medium (RPMI1640), 12.5 mL of fetal calf serum (FCS), and 100 μL of Gentamycin. After two to three hours of incubation at 36.5°C, the infected mouse was euthanized with $CO_2$, and a heart puncture was used to collect the blood. Subsequently, 2.5 mL of the schizont culture

**Table 2. List of the primers.**

| Primer Name | Orientation | Sequences | Binding Position (S1 Fig) |
|---|---|---|---|
| ::yfcu | Forward | GTGACAGGGGGAATG | 1 |
| | Reverse | GATAGCACTACCACCGG | 2 |
| mTagBFP2 | Forward | ggatccATGAGCGAGCTGATTAAGGAG | 3 |
| | Reverse | gcggccgcTTAATTAAGCTTGTGCCCCAGTT | 4 |
| | Reverse2 | *GATGAAGGTCTTGCTGCCGTAG* | 5 |
| eBFP2 | Forward | ggatccATGGTGAGCAAGGGCGAGGAG | 6 |
| | Reverse | gcggccgcCTACTTGTACAGCTCGTCC | 7 |
| | Reverse2 | *CTTCATGTGGTCGGGGTAGC* | 8 |
| 230p locus | Forward | GCAAAGTGAAGTTCAAATATGTG | 9 |
| | Reverse | GTGACTTTCAGTGAAATCGC | 10 |
| α-Tub | Forward | TGGAGCAGGAAATAACTGGG | |
| | Reverse | ACCTGACATAGCGGCTGAAA | |

were transferred into a Falcon tube. The blood-medium suspension was centrifuged for 8 minutes at 450 g. The supernatant was removed under the hood, and the pellet was resuspended with 2 mL of warm Schizont culture medium. The suspension was then transferred carefully and slowly into a schizont culture flask with 5% $CO_2$, 5% $O_2$ and 90% $N_2$ for 90 seconds. The culture was then incubated with slow rotation overnight at 36.5°C, as described previously [35].

The next day, 10 mL of a 60% Nycodenz-PBS mixture (27.7 mL Nycodenz and 22.5 mL 1 x PBS) were carefully pipetted below the schizont culture. The Falcon tubes were then centrifuged for 20 minutes at 450 g without acceleration or breaks. After centrifugation, a gradient was observed (top layer: schizont medium, middle layer: schizonts, lower layer: Nycodenz-PBS mix, bottom: pellet of RBCs and other parasite stages). Using a pipette, the schizont layer was carefully removed and transferred into a new Falcon tube. After an additional centrifugation at 11 000 g without acceleration or breaks, the cells were resuspended in transfection mix composed of 5 µg of digested DNA construct (*5'hsp70-eBFP2-3'hsp70 and 5'hsp70-mTagBFP-3'hsp70*) were filled up to 100 µL with Nucleofector transfection buffer from the Amaxaⓡ Human T cell Nucleofectorⓡ Kit. The solution was pipetted into an electroporation cuvette and placed into the Amaxa Nucleofector device. After electroporation using the U33 program, the solution was immediately transferred to an Eppendorf tube containing 100 µL of RPMI1640 without serum. 200 µL of the transfected schizonts were IV injected into naive mice.

The BFP constructs were integrated into the silent 230P genomic locus of the modified *P. berghei* GIMO mother line. The 230P locus contains a fusion gene of the following selectable markers: human dihydrofolate reductase, yeast cytosine deaminase, and uridyl phosphoribosyl transferase (hdhfr::yfcu). Double-crossover homologous recombination was used to replace the selectable marker cassettes at the target locations to introduce the BFP-containing constructs. Negative selection was then carried out a day after the transfection with 5-fluorocytosine (Abcam, ab141197) in drinking water to select for the BFP transgenic parasites that have the BFP-reporter cassette introduced into the 230p locus and the hdhfr::yfcu marker removed (S1 Fig). The correct integration of the BFP constructs into the *P. berghei* genome was further confirmed by integration PCR (S1 Fig, S1 Raw images). The primers used and their specific binding positions are listed in Table 2 and S1 Fig.

## Isolation of eBFP2 transgenic line

To generate the transgenic eBFP2 parasite line, a limiting dilution was carried out as described [36, 37]. Briefly, a naive mouse was pre-infected with the transfected BFP *P. berghei* parasite

lines. When the parasitemia reached 0.2–1%, around 2 μL of blood was collected from the tail vein of the pre-infected mouse and put into an Eppendorf tube with 1 mL of RPMI1640. The tube was gently inverted several times, and the blood mixture was transferred to a Falcon tube with 9 mL of RPMI 1640. Afterwards, 10 μL were taken from the solution, and the erythrocytes were counted with a Neubauer cell counting chamber. The volume and the dilution factor for limiting injection were determined by calculating the average number of erythrocytes per blood parasite. A dilution of 1:100 in RPMI1640 was made. Per mouse, 200 μL of the final dilution was IV-injected using a 31G insulin syringe. A total of ten pre-warmed naive mice were infected, each with an average of one single parasite. The parasitemia of the infected mice was checked every day for seven days post-infection. The positive mice were euthanized with $CO_2$, and the blood was collected by heart puncture, from which the genomic DNA was isolated, and subsequently, the parasite lines were confirmed by integration PCR.

## Oocyst size and volume measurements

To assess the performance of the transgenic BFP parasite lines in the sexual mosquito stage, naive mice were pre-infected and passaged as described above. Three days post-passage, the parasitemia was checked, and when the gametocytes reached 0.75 to 1%, the infected mice were anaesthetized and used to feed *Anopheles stephensi* mosquitoes as previously described [38, 39]. At days 5, 7, 11, and 14 after feeding, ten mosquitoes were dissected and their midguts were collected in 1 x PBS, placed on a microscopy slide coated with 21-wells, and covered with a coverslip. Subsequently, they were imaged using the 2.5x and 63x oil objectives on a Leica DM6000B, equipped with an EL6000 lamp. The number of oocysts was accessed using the find maxima plugin in Fiji, and the oocyst volumes were calculated from area measurements in Fiji using the formula $V = 4/3\pi r^3$, where V represents the volume and r is the radius [40]. The experiments were carried out in triplicate for each of the parasite lines.

## Sporozoite motility assay

The sporozoite motility was measured 18 days post-feeding when the sporozoites were supposed to be released from the oocysts into the salivary glands of the infected mosquitoes as previously described [41]. In total, ten mosquitoes were dissected, and the salivary glands were collected in Iscove's Modified Dulbecco's Media (IMDM). The sporozoites got activated with Minimum Essential Media (MEM) (BioConcept, 1-31F01-I) supplemented with Earle's salts and 10% heat-inactivated fetal calf serum (FCS, GE Healthcare), referred to as the completed medium. This solution was centrifuged for 5 min at 11'000 g, the supernatant was discarded, and the pellet was resuspended with 40 μL MEM. 20 μL of this solution were transferred into a 96-well glass bottom plate (Cellvis). The plate was spun down for 5 minutes at 200 g and finally incubated for 10 min at 37°C before being imaged using an inverted Leica DM5500B microscope equipped with an EL6000 lamp. Maximum-intensity z-projections were used to visualize the sporozoite motility pathways in Fiji. Furthermore, the sporozoite velocity was measured using the MTrack2 plugin from Nico Stuurman [40, 42]. After processing the image sequence into a binary image, this plugin measured the sporozoite speed by tracking the sporozoite tip frame by frame to calculate the approximate trajectory during a given time range.

## Immunofluorescence assay (IFA) of fixed HeLa cells

To access the development of the BFP-expressing parasites in the mammalian cells, 40'000 HeLa cells were seeded into a 24-well plate on a coverslip containing MEM-completed medium. The next day, the cells were infected with 20'000 sporozoites. At specific hours post-infection (6 hpi, 24 hpi, 30 hpi, 48 hpi, and 56 hpi), the cells were fixed with 4%

paraformaldehyde (PFA) in PBS for 10 min at room temperature as previously described [43]. After two washes with 1 x PBS, the cells were blocked with 10% FCS in 1 x PBS for 20 min at room temperature. Subsequently, the cells were incubated with the first antibodies diluted in 10% FCS for 1 hour at room temperature. The primary antibodies used for the cells infected with the BFP-expressing parasites were Rabbit anti-UIS4 (1:1'000, P-Sinnis, Baltimore) and Mouse anti-LAMP1 (1:1'000, Developmental Hybridoma Bank, clone H4A3, 1:1'000) while Rabbit anti-LAMP1 (1:1000, Cell Signaling, CS9091P) and Chicken anti-UIS4 (1:500, Proteogenix) were used for the cell infected with the mCherry-expressing parasites. After another washing step, the secondary antibodies, anti-Rabbit Cy5 (1:1'000, Jackson Immuno Research), anti-Mouse Alexa 594 (1:1'000, Invitrogen Molecular Probes, A11032), and anti-chicken Cy5 (1:1'000, Jackson Immuno Research) and anti-rabbit Alexa 350 (1:1'000, Invitrogen Molecular Probes, A11069), were added to the BFP and mCherry expressing parasite-infected cells, respectively. Single-stain controls have been performed to ensure that bleed-through was negligible. Incubation was carried out protected from light for 1 hour before staining the actin polymerization with phalloidin Alexa Fluor 488 (1:1'000, Invitrogen Molecular Probes, A123779). Finally, the coverslips were mounted with 5 μL ProLong™ Gold Antifade Mounting (Invitrogen, P36930) and imaged using the 60X objective of a Nikon Crest V3 spinning disc microscope in widefield mode.

The host lysosome interaction with the parasite PVM was determined using the 3D and 4D image analysis software Imaris (Imaris X64, 9.9.0 [March 11, 2022]). In order to measure the number of lysosomes attached to the parasite PVM (0 to 0.5 μm) in infected HeLa cells, the PVM stained with anti-UIS4 was used to compute the isosurface, and the lysosomal vesicles stained with anti-LAMP1 were used to compute isospots as previously described [38].

## Parasite size measurement and detached cell formation

The parasite size in the HeLa cells was measured as previously described [2, 44]. Briefly, 40'000 HeLa cells per well were seeded in triplicate in 96-well plates. The next day, each replicate was infected with 20'000 sporozoites from the different parasite lines. After 2 hours of incubation at 37°C, the cells from one well were seeded into 8 wells of two 96-well plates (i.e., 4 wells per plate). At specific time points post-infection (6 hpi, 24 hpi, 48 hpi, and 56 hpi), one 96-well plate was sequentially imaged with the 10x objective of an InCell Analyzer 2000 microscope. Analysis using the InCell Developer tool allowed individual parasites to be separated from the background by defining parameters such as kernel size, sensitivity, and the range for parasite size. Based on the size of each parasite, parasite growth patterns were measured. The absolute number of parasites at specific time points was used to calculate the percentage of parasite survival over time (6 hpi = 100%). To measure the formation of detached cells, they (together with thousands of merozoites inside of them) were harvested 65 hpi from the second plate by gently pipetting the supernatant of each well up and down three times. Finally, the solution containing the detached cells was transferred to another 96-well plate, and only the mature detached cells were counted under the microscope. The number of detached cells per triplicate was divided by the number of parasites present at 48 hpi to obtain the detached cell formation rate in percentage.

## Live cell imaging (movie) of the parasite development in HeLa cells

For live cell imaging of parasite development in HeLa cells, the cells were infected as described above. After 2 h of incubation at 37°C, they were reseeded on a dark 24-well glass bottom plate (Cellvis) to image the cells under a 40X objective of Nikon W1 LIPSI spinning disk microscope or 63X objective of a DMI 6000B Leica inverted microscope. Using the "Mark and find" function of the LEICA software, each invaded parasite could be marked and pictured every hour to

finally simulate a movie showing the parasite's development within the living HeLa cell over time, as described previously [19].

## Flow cytometry

To measure the parasitemia ratio of the infected mice, a few drops (around 2 μl) of blood from the tail vein of the mouse were added to 500 μL of 1 x PBS and measured by flow cytometry using the MoFlo ASTRIOS EQ as previously described [45]. mCherry (Excitation 561–614) and BFP (Excitation 405–450) lasers were used to respectively select the mCherry and BFP transgenic expressing parasites in infected erythrocytes. The data were analyzed using the NanoSight NS300 and FCS Express Flow Cytometry Analysis Software (version 7.22.0006).

## Data analysis

The data were analyzed with GraphPad Prism version 9.0.0 for Mac, GraphPad Software, San Diego, California, USA (www.graphpad.com). The experiments were carried out in triplicate. The statistical comparison tests were carried out using one-way ANOVA with Tukey's multiple comparisons test (*$p \leq 0.05$, **$p \leq 0.01$, ***$p \leq 0.001$). The microscope images were analyzed using FIJI software. When specified, the images were deconvolved with Huygens Professional version [Huygens Remote Manager v3.8] (Scientific Volume Imaging, The Netherlands).

# Results

## Generation of the eBFP2 and mTagBFP2 transgenic parasite lines

To generate the two cytosolic BFP-expressing *P. berghei* parasite lines, the eBFP2 and mTagBF-P2-reporter cassettes were individually transfected into the GIMO acceptor 1596 parasite mother line [29, 30]. In this process, constructs containing the *5'hsp70-eBFP2-3'hsp70 and 5'hsp70-mTagBFP2-3'hsp70* cassettes were integrated via double-crossover homologous recombination into the modified *P. berghei 230p* locus, which already expressed the *hdhfr:yfcu* selectable marker cassette [34]. This selectable maker cassette facilitated enrichment of the PbeBFP2- or PbmTagBFP2-transfected parasites by administering the pro-drug 5-fluorocytosine as a negative selection agent (S1A Fig). The *PbmTagBFP2*-infected erythrocytes were selectively isolated from a mixed population by FACS (S1B Fig). Conversely, the *PbeBFP2* transgenic parasite lines were isolated through limiting dilution (S1C Fig). Out of ten initially infected mice, six tested positive, and integration PCR confirmed the transgenic populations for four out of these six mice. Integration PCRs targeting the 5' and 3' UTR as well as the entire gene construct were also performed on the mTagBFP2 transgenic parasite line, verifying the correct integration of the constructs in the genome (S1D and S1E Fig). Following the successful generation of these two transgenic reporter lines, we verified the BFP visibility during blood stage development. In both parasite lines the BFP was clearly visible by fluorescence microscopy in all stages in blood: ring (R), trophozoite (T), gametocyte (G), and schizont (Sc) stages (Fig 1A and 1B). The mean fluorescent intensity of the BFP signals were compared by flow cytometry (Fig 1C and 1D) and revealed that mTagBFP was about 4.5 times stronger than in eBFP2 parasites (Fig 1E and 1F).

## Investigation of the PbeBFP2 and PbmTagBFP2 reporter parasite lines development within the mosquito midgut

The goal of this study was to create BFP-expressing parasite lines capable of normal development throughout their entire life cycle, spanning from the sexual stage in mosquitoes to the

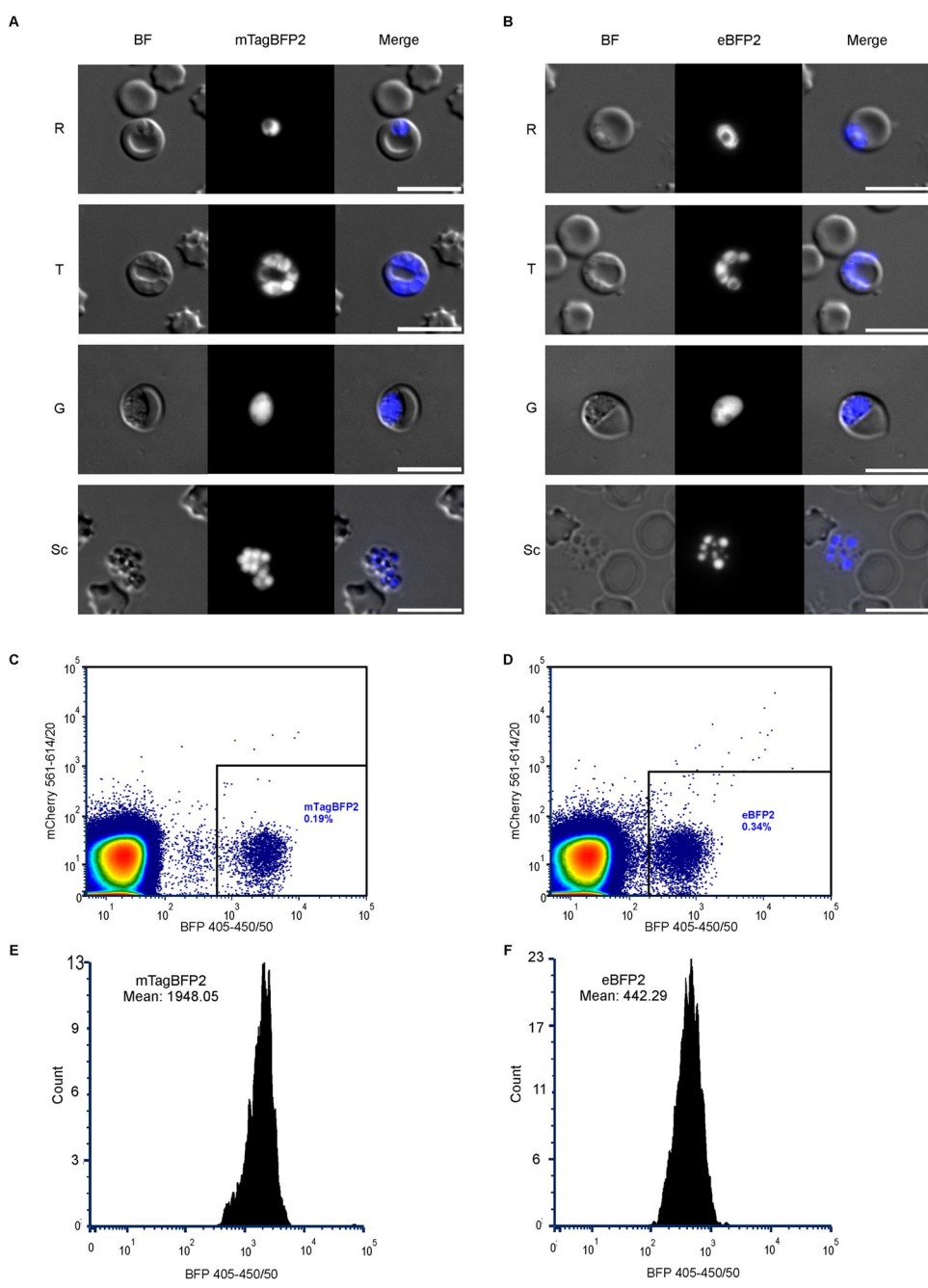

**Fig 1. mTagBFP2 and eBFP2 transgenic parasite were visible during blood stage development. (A)** and **(B)** display widefield microscope images showing the development of the PbmTagBFP2 and PbeBFP2 parasite lines in erythrocytes, highlighting ring (R), trophozoite (T), gametocyte (G), and schizont (Sc) stages of the BFP-expressing parasites. Scale bars = 10 μm. **(C)** Flow cytometry profile of PbmTagBFP2-infected mice indicating a parasitemia of 0.19%. **(D)** Flow cytometry profile of PbeBFP2-infected mice showing a parasitemia of 0.34%. (E) and (F) depict the intensity histograms for PbmTagBFP2 and PbeBFP2, respectively. The mean fluorescence intensity for PbmTagBFP2 is 1948.05 a.u., while for PbeBFP2 it is 442.29 a.u.

asexual stages in the liver and blood. To achieve this, both BFP-expressing parasite lines and as a control mCherry-expressing parasites [30] were fed to female *Anopheles* mosquitoes. To assess parasite development within the mosquito midgut at 7 and 14 days post-feeding, ten

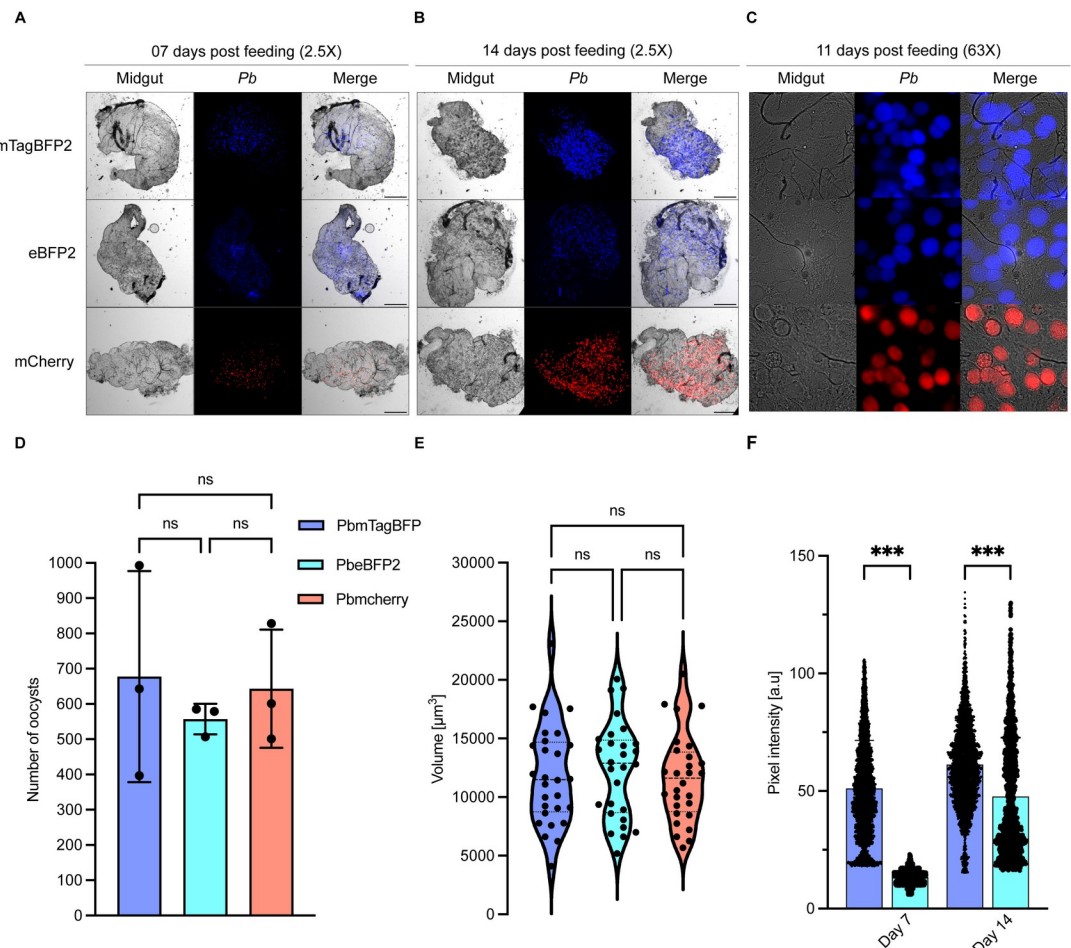

**Fig 2. mTagBFP2 and eBFP2 transgenic parasite development in the mosquito midgut.** **(A)** and **(B)** display images of mosquito midguts containing oocysts expressing BFP or mCherry 7 and 14 days post-feed (acquired using a 2.5x widefield microscope objective). The images of mosquito midguts containing oocysts in **(C)** were imaged eleven days post-feed with a 63x objective widefield microscope. **(D)** At day 7 post-feed, the total number of oocysts per midgut was computed using the "find maxima" plugin in Fiji. The mean oocyst number in triplicate was 680, 560, and 670, respectively, for the PbmTagBFP2, PbeBFP2, and PbmCherry parasite lines, with no significant differences. **(E)** shows the volume of the oocysts at eleven days post-feed. The average oocyst volume was between 11000 and 13000 $\mu m^3$ for all the parasite lines, with no significant differences. **(F)** represents the pixel intensity (a.u., arbitrary units) of the oocysts at seven and eleven days post-feeding. At day 7 post-feed, pixel intensity averages were 51 and 13.3, respectively, for the PbmTagBFP2 and PbeBFP2, and 61.2 and 47.7, respectively, for the PbmTagBFP2, and PbeBFP2 at day 14 post-feed. **(D)**, **(E)**, and **(F)** were color-coded: blue = PbmTagBFP; cyan = PbeBFP2; and red = PbmCherry. The comparisons of oocyst number, volume, and pixel intensity between the parasite lines were conducted using one-way analysis of variance (ANOVA) with Tukey's multiple comparisons test (*p $\leq$ 0.05, **p $\leq$ 0.01, ***p $\leq$ 0.001). Image scale bars: **(A)** and **(B)** = 500 $\mu m$; **(C)** = 10 $\mu m$.

midguts were dissected and examined using low magnification (2.5X objective) widefield microscopy (Fig 2A and 2B), and the oocyst number was quantified at 7days post-feeding (Fig 2D). The PbeBFP2 and PbmTagBFP2 reporter parasite lines exhibited oocyst counts comparable to the PbmCherry control parasites. Typically, *Plasmodium* oocysts are oval or spherical [46]. To determine if BFP-expressing parasites maintained this characteristic oocyst form, we analyzed oocyst morphology at 11 days post-feeding (Fig 2C). Additionally, the volume of individual oocysts was measured (Fig 2E). No significant differences were observed in the size or morphology of oocysts between the control and the BFP-expressing parasites. The intensity of the BFP reporter expression under the Hsp70 promoter within the mosquito midgut was

also evaluated by measuring the pixel intensity of oocysts intensities of both BFP-expressing parasites (Fig 2F). The PbmTagBFP2 oocysts displayed the brightest fluorescent signals compared to those of PbeBFP2 at 7 and 11 post-feeding (Fig 2F). Despite this, these findings confirm the normal progression of the BFP-expressing parasites throughout the mosquito stage.

## Quantification of the PbeBFP2 and PbmTagBFP2 sporozoite motility and exo-erythrocytic form development

During *Plasmodium* development, liver infection is a critical bottleneck in the parasite's life cycle [47]. The successful invasion of hepatocytes is dependent on the gliding motility of the sporozoites [48, 49]. To assess the motility of the sporozoites from the newly generated parasite lines, salivary gland sporozoites (day 18 post-feeding) from infected mosquitoes were placed on glass bottom dishes and imaged by wide-field microscopy. Both the movement and velocity of the sporozoites were quantified using live cell imaging (S1–S3 Movies). The BFP-expressing parasite lines exhibited a normal circular movement pattern similar to that observed in the PbmCherry parasites (Fig 3A, S1 to S3 Movies). The average velocity recorded was approximately 1.5 μm/s (Fig 3B) and is consistent with previous findings [48–51]. Following the assessment of gliding motility, the development of the parasites into exo-erythrocytic forms within HeLa cells was evaluated at multiple post-infection time points (6, 24, 30, 48, and 56 hpi). After fixing the cells, an immunofluorescence assay (IFA) was conducted by staining polymerized host cell actin with phalloidin to identify and delineate the border of individual cells. A hallmark of normal infection, the formation of a parasitophorous vacuole membrane (PVM), was visualized using IFA with antibodies targeting the parasite protein upregulated in infective sporozoite gene 4 (UIS4). As depicted in Fig 3C–3E, both BFP-expressing parasite lines demonstrated prominent UIS4 staining, akin to the mCherry control line, indicating typical infection patterns. Furthermore, live cell imaging of infected HeLa cells demonstrated normal liver stage development of the parasites (S4–S8 Movies). To evaluate the expression level of the two BFP reporters under the Hsp70 promoter during liver stage development, the pixel intensities of live cell images were measured at 48 hpi (Fig 3F). PbmTagBFP2 parasites showed significantly higher brightness than the PbeBFP2. Additionally, parasite size and survival were quantitatively assessed in host cells at various post-infection intervals. No significant differences were noted in size or survival between the BFP-expressing parasite lines and the PbmCherry line (Fig 4A and 4B). Finally, to determine if the two BFP-expressing parasite lines could fully complete the hepatic stage in vitro, detached cell formation rates and the parasites survival were determined at 65 hpi. No significant differences were observed between the BPF-expressing lines and control parasites (Fig 4C). These results demonstrate that the PbeBFP2 and PbmTagBFP2 parasite lines develop comparably to the PbmCherry control line throughout the *P. berghei* liver stage.

## Interaction of host cell lysosomes with intracellular PbeBFP2 and PbmTagBFP2 parasites during *Plasmodium* hepatic stage development

Recent observations have shown that lysosomes are recruited to the parasite parasitophorous vacuole membrane (PVM), as evidenced by the incorporation of Lysosomal-associated membrane protein 1 (LAMP1) into the PVM [9, 38, 52]. To verify whether the two BFP-expressing parasite lines similarly recruit host cell lysosomes to the PVM as observed with the PbmCherry control line, parasite-infected HeLa cells were fixed and labeled with antibodies detecting LAMP1 and the PVM marker UIS4 (S2A–S2C Fig). Our analysis revealed that approximately 40% of the host cell lysosomes are located within a close proximity of less than 0.5 μm to the

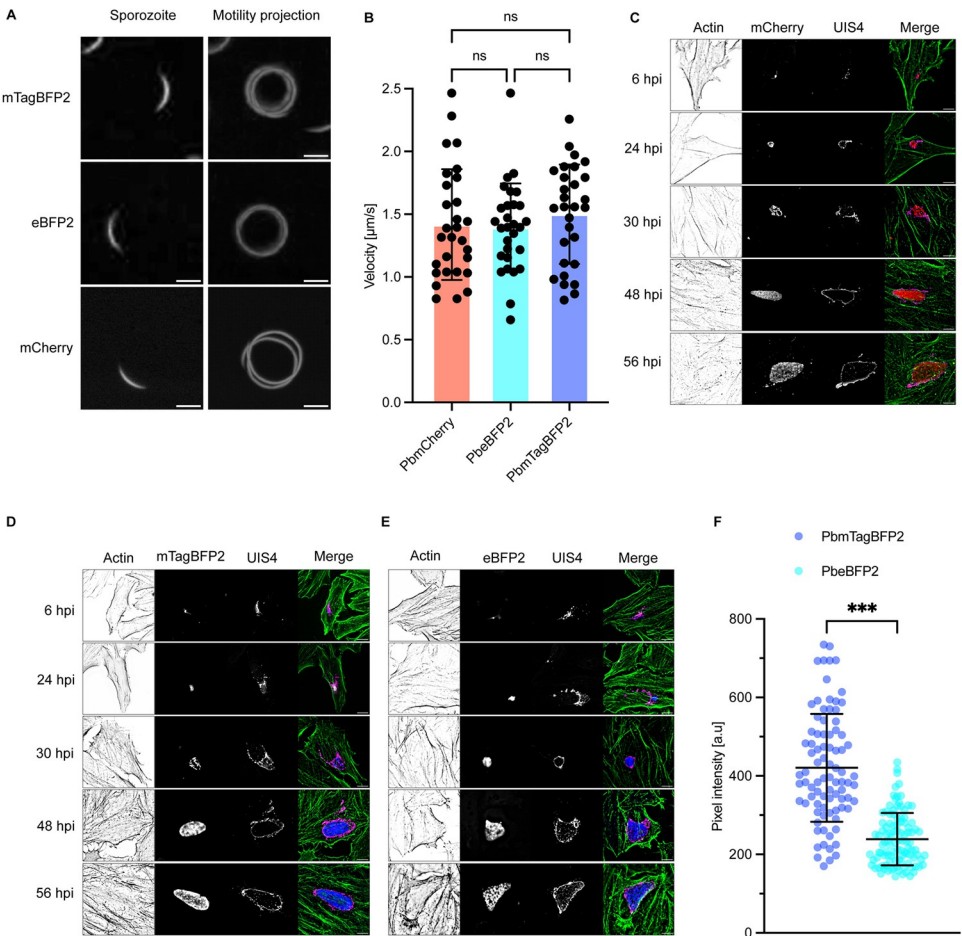

**Fig 3. The BFP transgenic parasites develop normally throughout the liver stage. (A)** illustrates the sporozoite motility projection. Transgenic sporozoites expressing BFP and mCherry were isolated from mosquito salivary glands and imaged live using a wide-field microscope. FIJI software was used to project the circular motility pathways of the parasites. **(B)** The velocities of the sporozoites were measured and compared between the BFP-expressing transgenic parasites and the PbmCherry control. The mean motility rates were 1.4, 1.38, and 1.5 μm/s for PbmCherry, PbeBFP2, and PbmTagBFP2, respectively. **(C)**, **(D)**, and **(E)** display images from an immunofluorescence assay of fixed parasite-infected HeLa cells, showing sporozoite infection at 6 hpi (hours post infection), trophozoite stages at 24 hpi, schizogony at 30 and 48 hpi, and the cytomere stage at 56 hpi. Polymerized host cell actin was stained with phalloidin dye conjugated to Alexa Fluor 488, and the parasite parasitophorous vacuole membrane (PVM) was stained with anti-UIS4 (upregulated in infective sporozoite gene 4) antiserum. **(F)** compares the BFP signal intensities of PbmTagBFP2 and PbeBFP2 parasites in live HeLa cells at 48 hpi, captured using confocal microscopy with a 40x objective. The average pixel intensity for PbmTagBFP2 was 420.65 a.u., while for PbeBFP2 it was 238.81 a.u. The bars were color-coded as indicated in the graphs (red: PbmCherry, blue: PbmTagBFP2, cyan: PbeBFP2). Mean pixel intensities were compared using a two-tailed t-test and mean velocities were compared using one-way ANOVA with Tukey's multiple comparisons test ($*p \leq 0.05$, $**p \leq 0.01$, $***p \leq 0.001$). All experiments were conducted in triplicate. Scale bars = 10μm.

PVM. No significant differences are observed between the BFP-expressing parasites and the PbmCherry line (S2D Fig).

## Transition from the hepatic to the blood stage is not affected in PbeBFP2 and PbmTagBFP2 parasites in vivo

To confirm that the development of the PbeBFP2 and PbmTagBFP2 parasites is not adversely affected in vivo, three mice per parasite line were intravenously injected with 5,000 salivary

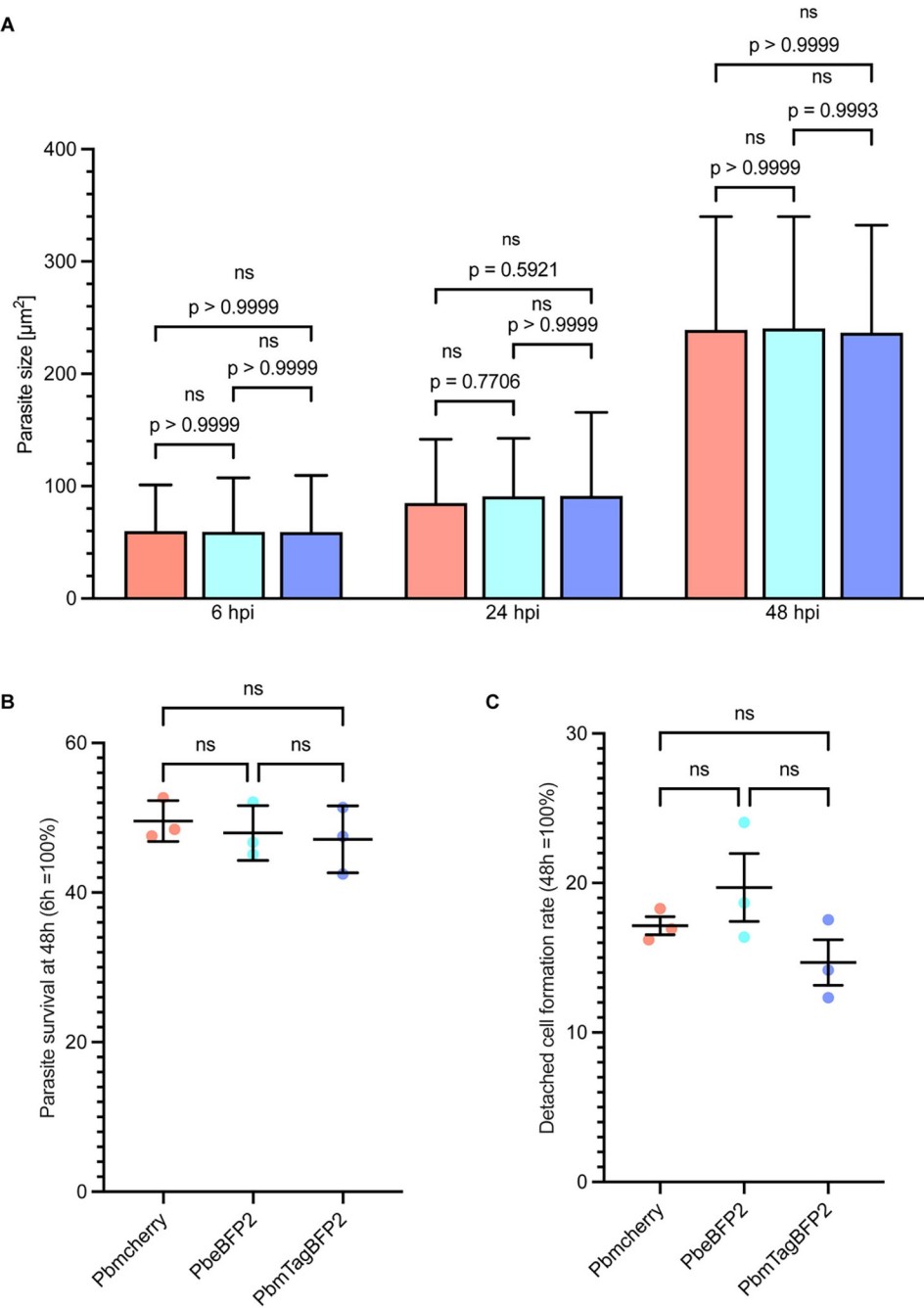

**Fig 4. Newly generated BFP parasite lines exhibit a normal exoerythrocytic development. (A)** represents the size measurement of liver stage parasites. 20000 sporozoites were used to infect HeLa cells. The sizes of both BFP transgenic parasites and PbmCherry were calculated in HeLa cells over a time course of 6, 24, and 48 hpi using one-way ANOVA with Tukey's multiple comparison test. The bar plots depict the mean size of the parasites along with their standard deviation. At 6 hpi, the mean size was approximately 50 μm2, increasing to 100 μm2 at 24 hpi, 200 μm2 at 48 hpi, and 300 μm2 at 56 hpi in the PbmCherry, PbeBFP2, and PbmTagBFP2 parasite lines with no significant difference. **(B)** shows the ratio of normalized survival data for parasites at 48 hpi. The survival ratio of the parasites was calculated based on the absolute number of infected parasites at 6 hpi. Approximately 50% of the PbmCherry and BFP transgenic parasites developed through the normal *Plasmodium* liver stage. Based on the survival ratio of parasites at 48 hpi, the ratio of detached cell formation was calculated at 65 hpi, as depicted in Panel **(C)**. No significant difference was observed in the detachment cell formation of the three transgenic parasites. On average, approximately 20% of the surviving parasites in the three transgenic parasite lines formed detached cells. Each experiment was carried out in triplicate. The statistical comparison tests were carried out using one-way ANOVA with Tukey's multiple comparisons test (*p ≤ 0.05, **p ≤ 0.01, ***p ≤ 0.001). Scale bars = 10 μm.

gland sporozoites each. For the co-infection, the mice were infected with 2500 sporozoites from each parasite clones. The progression of the parasite's blood-stage infection was monitored daily from the day of infection by flow cytometry (Fig 5A, 5C and 5E) until day 5 post infection (Fig 5B, 5D and 5F). All mice were tested clearly positive from day 3 onwards, without significant growth difference between the two BFP-expressing lines and the PbmCherry control line, confirming the full viability of the new parasite lines (Fig 5B). Coinfections of BFP-expressing sporozoites together with the control sporozoites showed no negative impact on either parasite line (Fig 5C-5F). Additionally, the PbmCherry parasites exhibited a higher brightness than the BFP-expressing parasites during the asexual blood stage development. With mean fluorescence intensities of 14116, PbmCherry parasites were 7.5 times brighter than PbmTagBFP2, and these in turn were also about 4.5 times brighter than PbeBFP2 parasites (Figs 1E, 1F and 5C and 5E). It is important to note that different lasers were used to detect the mCherry and BFP fluorescent signals and that a fluorescent protein's brightness can also be affected by factors such as the laser power and sensitivity.

## Discussion

In this study, we describe the generation of blue fluorescent *P. berghei* reporter parasite lines that expand the toolkit available for investigating parasite biology. The introduction of BFPs into *P. berghei* was motivated by their utility as markers for cellular processes, providing an alternative to the traditionally used red and green fluorescent proteins. Among the two BFPs used, eBFP2 requires dimerization for fluorescence, while mTagBFP2 is inherently monomeric, making it more adaptable for imaging techniques that require single-molecule stability and broad compatibility. This distinction is especially significant in applications involving diverse molecular environments. Importantly, many fluorescent proteins, such as GFPs, show reduced fluorescence under acidic conditions, posing challenges for studies involving acidic organelles [53–57]. The low pKa of mTagBFP2 (2.7) enhances its photostability and usefulness in studying interactions within these compartments [16, 26–28].

We validated the expression of both mTagBFP2 and eBFP2 under the constitutive Hsp70 promoter, known for providing reliable expression across *P. berghei* life stages [58]. The Hsp70 family plays crucial roles in protein folding, a function that supports the consistent expression needed for reporter studies [21]. Although PbmCherry displayed higher overall brightness compared to the BFP-expressing lines (Fig 5D and 5E), mTagBFP2 was notably 4.5 times brighter than eBFP2, consistent with prior findings [16, 26, 27]. The observed differences in apparent brightness may be influenced by factors such as spectral properties, environmental conditions, and imaging system characteristics, including laser power and sensitivity.

Both BFP-expressing lines maintained strong, detectable fluorescence throughout the parasite's life cycle, forming normally sized and viable oocysts and successfully infecting HeLa cells in vitro and hepatocytes in vivo. Initial observations showed that PbeBFP2 and PbmTagBFP2 parasites exhibited lower fluorescence at early stages in HeLa cells; however, this was rectified with adjusted exposure settings. Additionally, the fluorescence of these reporters persisted post-fixation, and commercially available BFP antibodies could be used to amplify their signal if needed. This enhancement capability is particularly advantageous for studies where signal strength is a limiting factor.

Our data further demonstrate that BFP- and mCherry-expressing lines develop comparably during simultaneous infections, highlighting their potential for multiplexed imaging applications. This is particularly valuable given the known challenges of fluorescent protein stability in acidic environments. The inherent stability of BFPs at low pH makes them well-suited for examining host-pathogen interactions, such as the recruitment of lysosomes to the

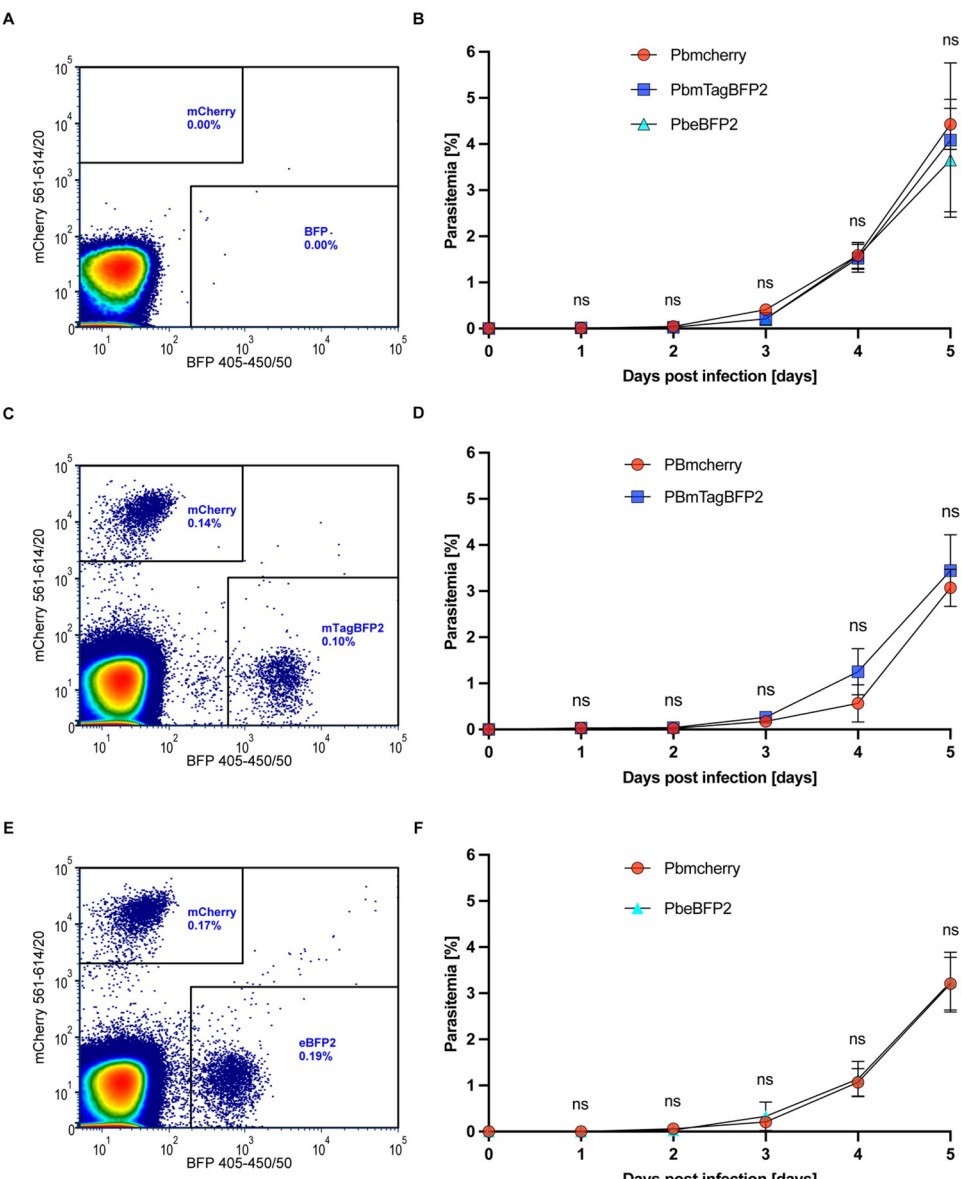

**Fig 5. Blood stage PbmTagBFP2 and PbeBFP2 parasite lines develop normally in vivo.** Development of BFP-expressing blood stage parasites in infected C57Bl/6 mice. **(A)** Flow cytometry profile of the blood from naive mice used as a control. The selection thresholds for the BFP (excitation 405–450) and mCherry (excitation 561–614) lasers are indicated by the outlined boxes. **(B)** The parasitemia of the infected mice was measured daily at the same hour. When the parasitemia reached between 4 and 5%, the mice were sacrificed. Each transgenic-line infection was carried out in triplicate. No significant difference was observed in the in vivo development from day 1 to 5 days post-infection of the BFP-expressing parasites compared to the PbmCherry control. **(C)** Flow cytometry profile of the blood from mice co-infected with PbmCherry and PbmTagBFP2 on day 3, showing parasitemia levels of 0.14% for PbmCherry and 0.10% for PbmTagBFP2. Due to the higher mean intensity of mTagBFP2 compared to eBFP2, the selection thresholds for mTagBFP2 in (C) were set higher than those for PbeBFP2 in (A) and (E). **(E)** shows the flow cytometry profile of the blood from mice co-infected with PbmCherry and PbeBFP2 on day 3, with parasitemia levels of 0.17% for PbmCherry and 0.19% for PbmTagBFP2. **(D)** and **(F)** represent the parasitemia progression in the blood of mice co-infected with PbmCherry and either PbeBFP2 or PbmTagBFP2 parasites. No significant differences were observed in the *in vivo* development from day 0 to 5 days post-infection between the co-infected BFP and PbmCherry-expressing parasites.

parasitophorous vacuole membrane (PVM), a process we confirmed in our early experiments [9, 59]. Nonetheless, additional studies are needed to fully characterize BFP performance under conditions of acidification and other variable intracellular environments.

Importantly, our results strongly suggest that blue fluorescent proteins would also be effective in *P. falciparum*, the causative agent of the most severe form of human malaria. The compatibility of BFPs with existing imaging techniques and their demonstrated stability in *P. berghei* provide a solid foundation for potential applications in *P. falciparum* studies. Implementing BFP-expressing reporter lines in *P. falciparum* could greatly enhance multiplexed imaging, enabling more detailed investigations of host-parasite interactions and organelle dynamics under conditions that closely mimic human infections.

In conclusion, the transgenic *P. berghei* parasites expressing BFPs under the Hsp70 promoter have demonstrated normal development throughout their life cycle, proving to be robust tools for advanced multiplexed imaging. These tools offer significant potential to enhance our understanding of host-pathogen interactions and cellular processes in malaria research. Further investigations are warranted to explore the full spectrum of applications and limitations, especially under challenging conditions like low pH, to maximize the utility of these fluorescent reporters.

## Supporting information

**S1 Fig. Generation of the transgenics blue fluorescent proteins expressing *P. berghei* parasites. (A)** showing the GIMO mother line PBANKA 230p 1596 locus, with the 3UTR and the 5' untranslated region (5' UTR) respectively flanked by the selection marker hdhfr::yfcu (black boxes). mTagBFP2 and eBFP2 were substituted for the yfcu selectable marker by homologous recombination under the Hsp70 promoter. Primers designed for the integration of PCR were indicated in the illustration and listed in Table 2. **(B)** Flow cytometry profile of the PbmTagBFP2 parasite when the parasitemia reached 0.23%. **(C)** Venn diagram displaying the limiting dilution of the PbeBFP2 parasite lines where 6 out of 10 mice tested positive. Integration PCR allowed for the identification of four of the six transgenic parasites that had no wild type. **(D)** Integration PCR results confirmed the proper integration of the mTagBFP2 and eBFP2 constructs. The expected size of the BFP fragment was determined to be 735 bp (solely in the TR), the yfcu fragment was 950 bp (exclusive to the WT), the 5' UTR (solely in the TR) at 1.5 kilobases, the 3' UTR (solely in the TR) at 3 kilobases, and the complete locus construct at 4.5kb. α-Tubulin was used as a control. TR = transgenics; WT = wild type, GIMO mother line (1596cl1); α-Tub = α-Tubulin.
(TIF)

**S2 Fig. Interaction of the mTagBFP2 and eBFP2 transgenic parasite with host cell lysosomes and late endosome. (A)**, **(B)**, and **(C)** immunofluorescent assays of fixed infected HeLa cells showing the host-cell lysosome vesicles interaction with the parasite PVM. HeLa cells were infected with sporozoites of the respective parasite lines and fixed at various time points (6, 24, 30, 48, and 56 hpi) during parasite development. Anti-UIS4 was used to stain the parasite PVM (magenta), anti-LAMP1 was used to stain the host cell lysosome (green), and phalloidin dye conjugated to Alexa Fluor 488 was used to stain the host cell polymerized actin (grey). The cells were imaged using a Nikon Crest V3 microscope in widefield mode. Z-stack images of the whole cell were acquired, and the host lysosome interaction with the parasite PVM was analyzed using the 3D and 4D image analysis software Imaris as described in the methods section. **(D)** Violin plot showing the percentage of host lysosomes (stained with anti-LAMP1) attached (0 μm) up to close proximity (0.5 μm) of the parasite PVM at 24 hpi. Neither of the BFP transgenic parasites differed significantly from the PbmCherry, with an average of 32%,

39%, and 46%, respectively, of PbmCherry, PbeBFP2, and PbmTagBFP2 LAMP1, present in the vicinity of the parasite PVM. Twenty infected cells from each transgenic parasite were analyzed in triplicate. The data were structured and analyzed using GraphPad Prism version 9.0.0 for Mac, GraphPad Software, San Diego, California, USA, www.graphpad.com. The statistical comparison tests were carried out using one-way ANOVA with Tukey's multiple comparisons test (*p ≤ 0.05, **p ≤ 0.01, ***p ≤ 0.001). The images were deconvolved with Huygens Professional version [Huygens Remote Manager v3.8] (Scientific Volume Imaging, The Netherlands, http://svi.nl). Scale bars = 10 μm.
(TIF)

**S1 Movie. PbmTagBFP2 sporozoite motility.**
(AVI)

**S2 Movie. PbeBFP2sporozoite motility.**
(AVI)

**S3 Movie. PbmCherry sporozoite motility.**
(AVI)

**S4 Movie. Live cell imaging of the PbmTagBFP2 liver stage development in infected HeLa cells using widefield microscopy from 6 hpi to 64 hpi.** The video highlights sporozoite infection at 6 hpi, the trophozoite stage at approximately 24 hpi, schizogony from 30 to 55 hpi, and the cytomere stage from 56 hpi onward.
(AVI)

**S5 Movie. Live cell imaging of PbmTagBFP2 liver stage development in infected HeLa cells captured using confocal microscopy from 25 hpi to 73hpi.** The video displays the trophozoite stage at around 25 hpi, schizogony stages from 30 to 54 hpi, the cytomere stage from 54 to 64 hpi, and detached cell formation with merozoites from 65 hpi to 73 hpi.
(AVI)

**S6 Movie. Live cell imaging of PbeBFP2 liver stage development in infected HeLa cells captured using widefield microscopy from 6 hpi to 63 hpi.** The video shows sporozoite infection at 6 hpi, the trophozoite stage at about 24 hpi, schizogony from 30 to 55 hpi, and the cytomere stage from 56 hpi.
(AVI)

**S7 Movie. Live cell imaging of the PbeBFP2 liver stage development in infected HeLa cells captured using confocal microscopy from 25 hpi to 73hpi.** The video illustrates the trophozoite stage at around 25 hpi, schizogony stages from 30 to 54 hpi, the cytomere stage from 54 to 64 hpi, and detached cell formation with merozoites from 65 hpi to 73 hpi.
(AVI)

**S8 Movie. Live cell imaging of the PbmCherry liver stage development in infected HeLa cells captured using widefield microscopy from 6 hpi to 63 hpi.** The video highlights sporozoite infection at 6 hpi, the trophozoite stage at about 24 hpi, schizogony from 30 to 55 hpi, and the cytomere stage from 56 hpi onwards.
(AVI)

**S1 Raw images. Original gel electrophoresis images.**
(PDF)

## Acknowledgments

The authors acknowledge Annina Bindschedler, Ruth Rehmann, and Bianca Manuela Berger for discussion, mosquitoes' maintenance, and assistance in revising the manuscript. We are grateful to Chris Janse for the pL1694 construct. The Microscopy Imaging Center (MIC) at the University of Bern in Switzerland provided support for the equipment used for microscopy. Kodzo Atchou acknowledges the Swiss Confederation for the awarded Government Excellence Scholarship.

## Author Contributions

**Conceptualization:** Kodzo Atchou, Reto Caldelari, Volker Heussler.

**Data curation:** Kodzo Atchou.

**Formal analysis:** Kodzo Atchou, Reto Caldelari.

**Funding acquisition:** Kodzo Atchou, Volker Heussler.

**Investigation:** Kodzo Atchou, Reto Caldelari, Magali Roques.

**Methodology:** Kodzo Atchou, Reto Caldelari, Magali Roques, Jacqueline Schmuckli-Maurer, Raphael Beyeler, Volker Heussler.

**Project administration:** Volker Heussler.

**Software:** Kodzo Atchou, Reto Caldelari.

**Supervision:** Volker Heussler.

**Validation:** Volker Heussler.

**Writing – original draft:** Kodzo Atchou.

**Writing – review & editing:** Kodzo Atchou, Reto Caldelari, Magali Roques, Jacqueline Schmuckli-Maurer, Raphael Beyeler, Volker Heussler.

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
