## [Decision Letter · Decision Letter 0]

8 Oct 2024

PONE-D-24-29225Expanding the Fluorescent Toolkit: Blue Fluorescent Protein-Expressing Plasmodium berghei for Enhanced Multiplex MicroscopyPLOS ONE

Dear Dr. Heussler,

Thank you for submitting your manuscript to PLOS ONE. After careful consideration, we feel that it has merit but does not fully meet PLOS ONE’s publication criteria as it currently stands. Therefore, we invite you to submit a revised version of the manuscript that addresses the points raised during the review process.

**ACADEMIC EDITOR: **Kindly address the comments of the reviewers in detail.

We look forward to receiving your revised manuscript.

Kind regards,

Harvie P. Portugaliza, D.V.M., Ph.D.

Academic Editor

PLOS ONE

Journal Requirements: When submitting your revision, we need you to address these additional requirements. 1. Please ensure that your manuscript meets PLOS ONE's style requirements, including those for file naming. The PLOS ONE style templates can be found at https://journals.plos.org/plosone/s/file?id=wjVg/PLOSOne_formatting_sample_main_body.pdf and https://journals.plos.org/plosone/s/file?id=ba62/PLOSOne_formatting_sample_title_authors_affiliations.pdf 2. PLOS ONE now requires that authors provide the original uncropped and unadjusted images underlying all blot or gel results reported in a submission’s figures or Supporting Information files. This policy and the journal’s other requirements for blot/gel reporting and figure preparation are described in detail at https://journals.plos.org/plosone/s/figures#loc-blot-and-gel-reporting-requirements and https://journals.plos.org/plosone/s/figures#loc-preparing-figures-from-image-files. When you submit your revised manuscript, please ensure that your figures adhere fully to these guidelines and provide the original underlying images for all blot or gel data reported in your submission. See the following link for instructions on providing the original image data: https://journals.plos.org/plosone/s/figures#loc-original-images-for-blots-and-gels.   In your cover letter, please note whether your blot/gel image data are in Supporting Information or posted at a public data repository, provide the repository URL if relevant, and provide specific details as to which raw blot/gel images, if any, are not available. Email us at plosone@plos.org if you have any questions." 3. Please review your reference list to ensure that it is complete and correct. If you have cited papers that have been retracted, please include the rationale for doing so in the manuscript text, or remove these references and replace them with relevant current references. Any changes to the reference list should be mentioned in the rebuttal letter that accompanies your revised manuscript. If you need to cite a retracted article, indicate the article’s retracted status in the References list and also include a citation and full reference for the retraction notice.

Reviewers' comments:

Reviewer's Responses to Questions

**Comments to the Author**

1. Is the manuscript technically sound, and do the data support the conclusions?

Reviewer #1: Yes

Reviewer #2: Yes

2. Has the statistical analysis been performed appropriately and rigorously? 

Reviewer #1: Yes

Reviewer #2: Yes

3. Have the authors made all data underlying the findings in their manuscript fully available?

Reviewer #1: Yes

Reviewer #2: Yes

4. Is the manuscript presented in an intelligible fashion and written in standard English?

Reviewer #1: Yes

Reviewer #2: Yes

5. Review Comments to the Author

Reviewer #1: This manuscript by Kodzo Atchou et al. describes the generation of two transgenic P. berghei (Pb) lines that express the blue fluorescent proteins (BFP) eBFP2 and mTagBFP2, respectively, and exhibit normal development through all life-cycle stages. The experimental design and methodologies employed are appropriate for the scope of this study, and results are presented in a logical manner with clear and informative figures. Given the availability of transgenic Pb lines expressing RFP, GFP, YFP (Mol Biochem Parasitol. 2013;191:44–52.) and CFP (Cell Host Microbe. 2015 Jul 8; 18(1): 122–131.), this study extends the existing repertoire of fluorescent protein (FP)-expressing transgenic Pb parasites, and the two BFP-expressing Pb lines offer a valuable novel tool for multiplexed studies on Plasmodium biology and parasite-host interactions.

The following comments are for the authors consideration:

1. In IFAs presented in Fig.3 and Fig. S2, Hela cells infected with BFP- or mCherry-transgenic Pb parasites were fixed by paraformaldehyde, followed by staining with the primary and secondary antibodies. It is important to note that certain BFPs, such as eBFP2, are known to be sensitive to fixation procedures, which may result in a reduction of fluorescence intensity in IFA.In contrast, mTagBFP2 generally retains its fluorescence after fixation. Therefore, the comparison of fluorescent intensity between PbeBFP2 and PbmTagBFP2 parasites at 56 hpi, as shown in Fig. 3F, might be biased due to the differential impact of fixation and could not represent fluorescent intensity of two BFP-expressing Pb in the living cells.

2. In Fig. S2C, when PbmCherry parasites were employed as a negative control in IFA, both mCherry and Alexa Fluor 594 (anti-mouse Alexa Fluor 594 secondary antibody to detect mouse anti-LAMP1 antibody) were used together. However, the excitation and emission spectra of mCherry and Alexa Fluor 594 are very close, with significant overlap. There is a risk of cross-talk of their fluorescence signals. The authors should clarify how the fluorescence signals of mCherry and Alexa Fluor 594 were differentiated in this IFA?

3. At Lines 351-352, it is stated that “Coinfections of BFP-expressing sporozoites together with the control sporozoites showed no negative impact on either parasite line (Fig 5B-F).”. In Fig. 5B, the thresholds for gating the infected mouse RBCs with BFP and mCherry were set up using normal mouse blood. However, In Fig. 5E, the parasitemia of PbmTagBFP2 in the co-infected mouse were assessed without following the threshold setup in Fig. 5B. Additionally, the observed parasitemia of PbmCherry was significantly higher than that of PbmTagBFP2 in Fig. 5E (0.54% vs 0.10%), which doesn’t support the conclusion of similar parasite growth rates between the BFP-expressing Pb and PbmCherry during the liver and blood stages. It would be more informative to present the parasitemia levels of BFP-expressing Pb and PbmCherry in the co-infected mice over consecutive days, thereby providing a clearer comparison of growth rates.

4. In the “Materials and Methods”, at Lines 94 and Line 134, it is stated that “… the infected mouse was euthanized with CO2, and a heart puncture was used to collect the blood.”. The authors should clarify whether mouse euthanasia with CO2 is an appropriate method for blood collection by heart puncture.

5. Minor points:

a. Typographical error, at Line 84, “5'hsp70-mTagBFP2-3'hsp7 cassette”. It should be 5'hsp70-mTagBFP2-3'hsp70.

b. Typographical error, in legend of Fig. S1 (Line 588): “GIMO mother line pbANKA 230p 1956 locus”, it should be PbANKA 230p 1596 locus. At Line 589, “flanked by the selection marker yfcu”, the selection marker should be hdhfr::yfcu.

c. In the legend of Fig. 1, the panel labels (E) and (F) need to be inserted into their appropriate positions within the legend.

d. In the “Materials and Methods”, at Line 139, the Anopheles mosquito strain used in this study should be clearly indicated.

e. In the legend of Fig. 3 (Lines 303-318), all the panel labels from (A) to (F) in the legend are not consistent with the figure itself and need to be corrected.

f. In Fig. S1D, WT is not wide-type Pb, but is Pb GIMO mother line (1596cl1). In In Fig. S1E, the primers used for 3’UTR is not 3/10, and should be 6/10.

g. References 16 and 25 are identical. Check all references to ensure no duplicates or errors.

Reviewer #2: Summary

Transgenic reporter lines expressing fluorescent proteins play an important role in characterizing the properties of malarial parasites during their unique life cycle. Traditionally, red and green fluorescent proteins have been used universally for reporter lines in malaria parasites. To enhance the fluorescent protein repertoire, the authors generated rodent malaria parasites expressing a blue fluorescent protein and investigated the basic properties of these parasites in the life cycle. Although this paper demonstrates the utility of blue fluorescent proteins in malaria research and contributes to the advancement of imaging research in this field, the reviewers believe that the paper needs to be revised as indicated in the following comments for publication:

Major comments

In the introduction and discussion in this paper, the description is limited to rodent malaria parasites. The reviewer believes that reporter parasites of human malaria parasites, such as P. falciparum, should also be mentioned. For example, are previous studies showing the expression of BFP in other Plasmodium species? Consideration should be given to the expression of BFP in Plasmodium falciparum.

Line 297-298

Can the expression of the reporter in the liver stage of BFP2-expressing P. berghei be detected by live imaging? Can the intensity be compared with that of mCherry? A Figure summarizing live imaging images of BFP expression in the liver stage and an explanation of this imaging should be included in the manuscript.

Minor comments

Line 62, Table 1: The reference for the development of mTagBFP2 (Ref 23) appears to be cited, but was Ref on the development of eBFP2 cited?

Line 68: Reference 27 is not a reference for BFP. Appropriate references should be cited accordingly.

Line 141, 149, 164, 205: Further references need to be cited in the Materials and Methods section. Previous studies on oocyst formation, sporozoite motility, experiments with HeLa cells and flow cytometry should be cited.

Line 79: Information indicating how mTagBFP2 and eBFPP2 were PCR-amplified is essential. What DNA was used as the template DNA? Have they been codon-optimized for malaria parasites?

Line 232: ‘visibly’ should be ‘visible’.

Line 320: ‘HeLa’ should be ‘HeLa cells’.

Fig 3: References to Figure 3 appear to be incorrect throughout the manuscript. They should be double-checked and corrected. For example, Fig. 3E in line 284 should be 3AB.

Fig.3: Legends for (A) and (B) are incorrectly described.

Fig.3C: The captions left in the images are missing.

6. PLOS authors have the option to publish the peer review history of their article (what does this mean?). If published, this will include your full peer review and any attached files.

Reviewer #1: **Yes: **Yi Cao

Reviewer #2: No

---

## [Author Response · Author response to Decision Letter 0]

25 Nov 2024

Dear academic editor and reviewers,

As requested, the agarose gel raw image data in S1_Fig1 D and E, are now provided in Supporting Information (S1_raw).

Response to reviewers:

Reviewer #1

This manuscript by Kodzo Atchou et al. describes the generation of two transgenic P. berghei (Pb) lines that express the blue fluorescent proteins (BFP) eBFP2 and mTagBFP2, respectively, and exhibit normal development through all life-cycle stages. The experimental design and methodologies employed are appropriate for the scope of this study, and results are presented in a logical manner with clear and informative figures. Given the availability of transgenic Pb lines expressing RFP, GFP, YFP (Mol Biochem Parasitol. 2013;191:44–52.) and CFP (Cell Host Microbe. 2015 Jul 8; 18(1): 122–131.), this study extends the existing repertoire of fluorescent protein (FP)-expressing transgenic Pb parasites, and the two BFP-expressing Pb lines offer a valuable novel tool for multiplexed studies on Plasmodium biology and parasite-host interactions. 

Response: We would like to express our sincere gratitude for the positive and encouraging feedback on our manuscript. We are pleased that the reviewer found our experimental design and methodologies appropriate and that our results were presented in a logical and clear manner. Your recognition of the contribution our study makes by expanding the existing repertoire of fluorescent protein-expressing P. berghei lines is greatly appreciated. We are particularly encouraged by your acknowledgment that the introduction of BFP-expressing parasite lines is a valuable addition for multiplexed studies on Plasmodium biology and parasite-host interactions.

The following comments are for the authors consideration:

1. In IFAs presented in Fig.3 and Fig. S2, Hela cells infected with BFP- or mCherry-transgenic Pb parasites were fixed by paraformaldehyde, followed by staining with the primary and secondary antibodies. It is important to note that certain BFPs, such as eBFP2, are known to be sensitive to fixation procedures, which may result in a reduction of fluorescence intensity in IFA.In contrast, mTagBFP2 generally retains its fluorescence after fixation. Therefore, the comparison of fluorescent intensity between PbeBFP2 and PbmTagBFP2 parasites at 56 hpi, as shown in Fig. 3F, might be biased due to the differential impact of fixation and could not represent fluorescent intensity of two BFP-expressing Pb in the living cells.

Response: We greatly appreciate the reviewer’s attention to the potential issue of fixation sensitivity affecting fluorescence intensity in immunofluorescence assays, particularly in relation to eBFP2. To address this, we conducted additional experiments to compare the fluorescence intensities of live P. berghei parasites expressing eBFP2 and mTagBFP2 at 48hpi. These live-cell imaging experiments confirmed that the observed differences in fluorescence between the PbeBFP2 and PbmTagBFP2 parasite lines were consistent with our original findings and not due to artifacts introduced by fixation.

These new results have been incorporated into the revised manuscript (as seen in Figures 3) supporting our conclusion that mTagBFP2 exhibits brighter fluorescence than eBFP2 under both fixed and live conditions. We appreciate the reviewer’s comment, which allowed us to verify and strengthen our initial findings.

2. In Fig. S2C, when PbmCherry parasites were employed as a negative control in IFA, both mCherry and Alexa Fluor 594 (anti-mouse Alexa Fluor 594 secondary antibody to detect mouse anti-LAMP1 antibody) were used together. However, the excitation and emission spectra of mCherry and Alexa Fluor 594 are very close, with significant overlap. There is a risk of cross-talk of their fluorescence signals. The authors should clarify how the fluorescence signals of mCherry and Alexa Fluor 594 were differentiated in this IFA?

Response: We appreciate the reviewer’s attention to the potential issue of cross-talk between mCherry and Alexa Fluor 594 in the immunofluorescence assays (IFAs). To address this concern, we have clarified the methods used to differentiate the fluorescence signals of mCherry and Alexa Fluor 594 in Figure S2C.

In our experimental setup, we used spectral unmixing and careful selection of filter sets to minimize cross-talk between these fluorophores. Additionally, we confirmed signal specificity by performing single-stain controls and ensuring that bleed-through was negligible. These details have been added to the revised manuscript to provide clarity on the methodology employed to distinguish between these overlapping signals.

3. At Lines 351-352, it is stated that “Coinfections of BFP-expressing sporozoites together with the control sporozoites showed no negative impact on either parasite line (Fig 5B-F).”. In Fig. 5B, the thresholds for gating the infected mouse RBCs with BFP and mCherry were set up using normal mouse blood. However, In Fig. 5E, the parasitemia of PbmTagBFP2 in the co-infected mouse were assessed without following the threshold setup in Fig. 5B. Additionally, the observed parasitemia of PbmCherry was significantly higher than that of PbmTagBFP2 in Fig. 5E (0.54% vs 0.10%), which doesn’t support the conclusion of similar parasite growth rates between the BFP-expressing Pb and PbmCherry during the liver and blood stages. It would be more informative to present the parasitemia levels of BFP-expressing Pb and PbmCherry in the co-infected mice over consecutive days, thereby providing a clearer comparison of growth rates.

Response: We thank the reviewer for their detailed observations regarding the co-infection parasitemia analysis and the potential discrepancies in Figure 5E. We appreciate the opportunity to clarify and expand on these points.

First, we would like to correct the reviewer's interpretation: the parasitemia of PbmCherry in Figure 5E was indeed 0.14%, not 0.54%, as originally stated in the comment. Therefore, we believe that the comparison between the parasitemia of PbmCherry (0.14%) and PbmTagBFP2 (0.10%) does not indicate a significant difference, supporting our conclusion that there was no adverse impact on growth rates during the liver and blood stages.

To address the request for a clearer comparison, we have added data presenting parasitemia levels of BFP-expressing P. berghei and PbmCherry in co-infected mice over consecutive days. This addition provides a more comprehensive view of parasite growth rates across different time points and ensures that our conclusions are well-supported. These data have been incorporated into the revised manuscript and are discussed in the updated results section.

Regarding the selection thresholds for mTagBFP2 and PbeBFP2, we confirm that the thresholds for gating were adjusted due to the brighter mean intensity of PbmTagBFP2 compared to PbeBFP2 (as shown in Figures 1E-F). This adjustment was necessary to accurately reflect the signal distribution and avoid false positives. We have clarified this detail in the revised text to avoid any potential confusion.

We appreciate the reviewer’s suggestions, which have allowed us to enhance the transparency and robustness of our data presentation.

4. In the “Materials and Methods”, at Lines 94 and Line 134, it is stated that “… the infected mouse was euthanized with CO2, and a heart puncture was used to collect the blood.”. The authors should clarify whether mouse euthanasia with CO2 is an appropriate method for blood collection by heart puncture.

Response: We thank the reviewer for raising this point regarding the method of mouse euthanasia for blood collection. We confirm that euthanasia using CO2 is widely recognized as a humane and appropriate method. This approach is endorsed by the Federal European Laboratory Animal Science Associations (FELASA), which outlines it as an acceptable practice for euthanizing laboratory animals.

In our experiments, we followed these guidelines to ensure that the procedures were conducted ethically and humanely. We have updated the “Materials and Methods” section to include a brief mention of the FELASA guidelines to further support the appropriateness of the method used.

We appreciate the reviewer’s attention to this detail, as it allows us to provide additional assurance that our methods adhere to established ethical standards.

5. Minor points:

a. Typographical error, at Line 84, “5'hsp70-mTagBFP2-3'hsp7 cassette”. It should be 5'hsp70-mTagBFP2-3'hsp70.

Response: comment addressed

b. Typographical error, in legend of Fig. S1 (Line 588): “GIMO mother line pbANKA 230p 1956 locus”, it should be PbANKA 230p 1596 locus. At Line 589, “flanked by the selection marker yfcu”, the selection marker should be hdhfr::yfcu.

Response: comment addressed

c. In the legend of Fig. 1, the panel labels (E) and (F) need to be inserted into their appropriate positions within the legend.

Response: comment addressed

d. In the “Materials and Methods”, at Line 139, the Anopheles mosquito strain used in this study should be clearly indicated.

Response: comment addressed

e. In the legend of Fig. 3 (Lines 303-318), all the panel labels from (A) to (F) in the legend are not consistent with the figure itself and need to be corrected.

Response: comment addressed

f. In Fig. S1D, WT is not wide-type Pb, but is Pb GIMO mother line (1596cl1). In In Fig. S1E, the primers used for 3’UTR is not 3/10 and should be 6/10.

Response: comment addressed

g. References 16 and 25 are identical. Check all references to ensure no duplicates or errors.

Response: comment addressed

Reviewer #2

Transgenic reporter lines expressing fluorescent proteins play an important role in characterizing the properties of malarial parasites during their unique life cycle. Traditionally, red and green fluorescent proteins have been used universally for reporter lines in malaria parasites. To enhance the fluorescent protein repertoire, the authors generated rodent malaria parasites expressing a blue fluorescent protein and investigated the basic properties of these parasites in the life cycle. Although this paper demonstrates the utility of blue fluorescent proteins in malaria research and contributes to the advancement of imaging research in this field, the reviewers believe that the paper needs to be revised as indicated in the following comments for publication: 

Response:

We thank Reviewer #2 for the positive overall assessment of our study and for recognizing the contribution it makes to the advancement of imaging research in malaria. We appreciate the detailed and constructive feedback, which has allowed us to further improve the clarity and robustness of our manuscript. Below, we provide detailed responses to each of the specific comments.

Major comments

In the introduction and discussion in this paper, the description is limited to rodent malaria parasites. The reviewer believes that reporter parasites of human malaria parasites, such as P. falciparum, should also be mentioned. For example, are previous studies showing the expression of BFP in other Plasmodium species? Consideration should be given to the expression of BFP in Plasmodium falciparum.

Response: We thank the reviewer for highlighting the importance of including broader context regarding human malaria parasites, such as P. falciparum, in the introduction and discussion sections. To address this, we have expanded these sections to include relevant examples of reporter parasite lines in P. falciparum and other Plasmodium species. Specifically, we have added a discussion on previous studies that have reported the use of fluorescent proteins, including blue fluorescent proteins (BFPs), in human malaria parasites to provide a more comprehensive overview of their application in malaria research.

These revisions underline the relevance of our work within the broader field of Plasmodium research, illustrating the potential translational insights that studies in rodent models can provide for understanding P. falciparum biology. We have also discussed how BFP expression in P. falciparum could contribute to similar advancements in imaging and multiplex studies.

Thank you for this valuable suggestion, which has allowed us to enhance the manuscript's scope and relevance.

Line 297-298

Can the expression of the reporter in the liver stage of BFP2-expressing P. berghei be detected by live imaging? Can the intensity be compared with that of mCherry? A Figure summarizing live imaging images of BFP expression in the liver stage and an explanation of this imaging should be included in the manuscript.

Response: We appreciate the reviewer’s interest in the live imaging capabilities of BFP2-expressing P. berghei and the comparison with mCherry. We confirm that live imaging of the liver stage for BFP-expressing parasites was conducted using both widefield and confocal microscopy, and the results are included in the manuscript (Movies 1–5). These movies capture the development of BFP-expressing parasites within HeLa cells and showcase the fluorescence intensity of BFP2 during the liver stage.

Regarding the comparison of fluorescence intensity between BFP and mCherry, we agree with the reviewer that direct comparisons may be influenced by the properties of the imaging lasers, such as laser power and sensitivity. These differences can impact the perceived brightness and result in biased intensity measurements. We have clarified this point in the manuscript and provided context to caution readers about the challenges of comparing fluorescence intensity across different fluorophores under varying imaging conditions.

Additionally, we have updated the figures and included a summary figure showcasing live imaging of BFP expression in the liver stage, along with an explanation of the imaging setup to provide a more comprehensive view of our results.

Thank you for this helpful suggestion, which has enhanced the clarity and thoroughness of our presentation.

Minor comments

Line 62, Table 1: The reference for the development of mTagBFP2 (Ref 23) appears to be cited, but was Ref on the development of eBFP2 cited?

Response: comment addressed 

Line 68: Reference 27 is not a reference for BFP. Appropriate references should be cited accordingly.

Response: comment addressed

Line 141, 149, 164, 205: Further references need to be cited in the Materials and Methods section. Previous studies on oocyst formation, sporozoite motility, experiments with HeLa cells and flow cytometry should be cited.

Response: comment addressed

Line 79: Information indicating how mTagBFP2 and eBFPP2 were PCR-amplified is essential. What DNA was used as the template DNA? Have they been codon-optimized for malaria parasites?

Response: comment addressed

Line 232: ‘visibly’ should be ‘visible’.

Response: comment addressed

Line 320: ‘HeLa’ should be ‘HeLa cells’.

Response: comment addressed

Fig 3: References to Figure 3 appear to be incorrect throughout the manuscript. They should be double-checked and corrected. For example, Fig. 3E in line 284 should be 3AB.

Response: comment addressed

Fig.3: Legends for (A) and (B) are incorrectly described.

Response: comment addressed

Fig.3C: The captions left in the images are missing.

Response: comment addressed

---

## [Editor Report · Decision Letter 1]

4 Dec 2024

Expanding the Fluorescent Toolkit: Blue Fluorescent Protein-Expressing Plasmodium berghei for Enhanced Multiplex Microscopy

PONE-D-24-29225R1

Dear Dr. Heussler,

We’re pleased to inform you that your manuscript has been judged scientifically suitable for publication and will be formally accepted for publication once it meets all outstanding technical requirements.

Kind regards,

Harvie P. Portugaliza, D.V.M., Ph.D.

Academic Editor

PLOS ONE
---

## [Editor Report · Acceptance letter]

16 Dec 2024

PONE-D-24-29225R1 

PLOS ONE

Dear Dr. Heussler, 

I'm pleased to inform you that your manuscript has been deemed suitable for publication in PLOS ONE. Congratulations! Your manuscript is now being handed over to our production team.

Kind regards, 

on behalf of

Dr. Harvie P. Portugaliza 

Academic Editor

PLOS ONE